# LONG-HORIZON VISUAL INSTRUCTION GENERATION WITH LOGIC AND ATTRIBUTE SELF-REFLECTION

**Yucheng Suo    Fan Ma    Kaixin Shen    Linchao Zhu    Yi Yang** *

ReLER Lab, CCAI, Zhejiang University, China

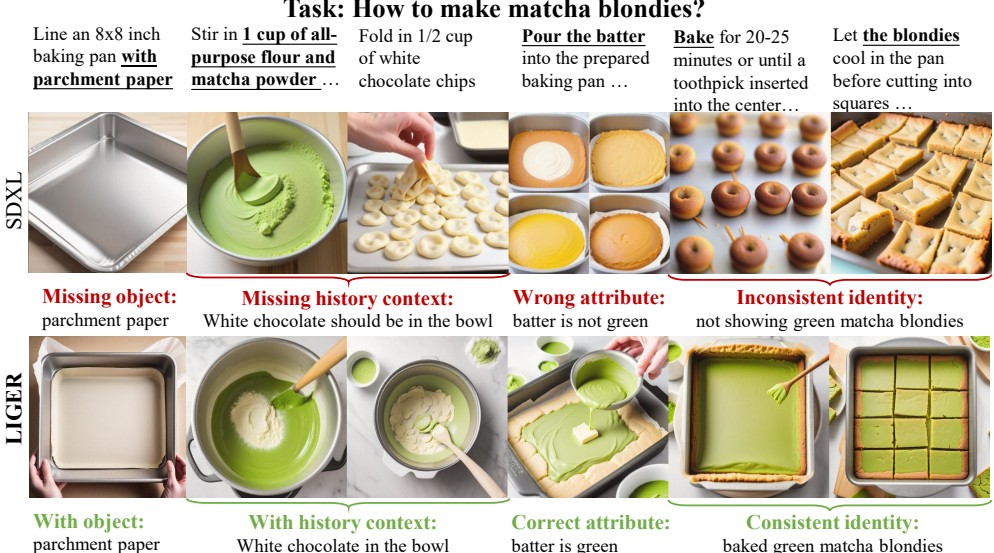

Figure 1: Visual instruction generated by LIGER, key merits are highlighted in the figure.

## ABSTRACT

Visual instructions for long-horizon tasks are crucial as they intuitively clarify complex concepts and enhance retention across extended steps. Directly generating a series of images using text-to-image models without considering the context of previous steps results in inconsistent images, increasing cognitive load. Additionally, the generated images often miss objects or the attributes such as color, shape, and state of the objects are inaccurate. To address these challenges, we propose **LIGER**, the first training-free framework for **L**ong-horizon **I**nstruction **GE**neration with logic and attribute self-**R**eflection. LIGER first generates a draft image for each step with the historical prompt and visual memory of previous steps. This step-by-step generation approach maintains consistency between images in long-horizon tasks. Moreover, LIGER utilizes various image editing tools to rectify errors including wrong attributes, logic errors, object redundancy, and identity inconsistency in the draft images. Through this self-reflection mechanism, LIGER improves the logic and object attribute correctness of the images. To verify whether the generated images assist human understanding, we manually curated a new benchmark consisting of various long-horizon tasks. Human-annotated ground truth expressions reflect the human-defined criteria for how an image should appear to be illustrative. Experiments demonstrate the visual instructions generated by LIGER are more comprehensive compared with baseline methods. Code and dataset are provided in `https://github.com/suoych/LIGER`.

---

*Corresponding Author.

# 1 INTRODUCTION

Humans learn to accomplish real-world tasks quickly through step-by-step text instructions. However, without visual aids, it is challenging to imagine the object attribute status and judge the completion status of the steps. For instance, when frying potato chips, merely reading the text description makes it hard to judge whether the chips are done. In contrast, viewing a video or a series of images accelerates individual understanding of task procedures, enhancing the success rate of completing various tasks. Generating illustrative visual instructions eases the comprehension burden and therefore becomes a crucial and trending task (Lu et al., 2023; Bordalo et al., 2024; Menon et al., 2024; Damen et al., 2024). Moreover, generating visual instructions unleashes the potential applications including multi-modal embodied agent perception and new task adaptation (Fan et al., 2024; Zhou et al., 2024a). In this paper, we aim to generate a series of images given task step descriptions.

A naive approach to generating visual instructions involves directly using text-to-image models, such as Latent Diffusion Models (LDMs) (Rombach et al., 2022). As Figure 1 illustrates, this method results in images lacking object consistency, thereby confusing users about the relationships between steps. To enhance image continuity, GenHowTo (Damen et al., 2024) trains a controllable U-Net (Ronneberger et al., 2015) model to enhance identity consistency. StackDiffusion (Menon et al., 2024) uses a diffusion model that takes concatenated latents from different steps as input. Sequential Latent Diffusion Model (SLDM) (Bordalo et al., 2024) trains a language model to regenerate consistent textual descriptions and use latents of the previous steps to enhance consistency. However, these approaches tend to produce overly consistent images that fail to capture changes in object states. An illustrative visual instruction should balance continuity with sufficient variability. This leads to the first challenge: the need for logical coherence across steps while allowing for appropriate changes.

Moreover, we empirically observe that the attributes of objects, *e.g.* color, state, and shape, might be incorrect in the images as depicted in Figure 1. These errors can accumulate, impacting the generation result of other steps and posing a significant challenge in long-horizon tasks. This leads to the second challenge, *i.e.* , attribute error and cumulation.

Our intuition for addressing these issues is to first generate a draft image for each step with the visual and textual context of previous steps, ensuring continuity between images. Then, through a process of self-reflection, we refine the draft images by adjusting for excessive continuity and correcting object attribute errors. This iterative approach not only prevents the accumulation of attribute errors in long-horizon tasks but also maintains appropriate logic relations across steps, similar to drafting and refining sketches.

To this end, we propose LIGER, a training-free framework for long-horizon visual instruction generation consisting of (1) historical prompt and visual memory, (2) self-reflection and memory calibration. Specifically, we leverage the reasoning ability of LLM to explicitly output history context for each step, facilitating relation comprehension. Inspired by the recent training-free identity consistent generation works (Zhou et al., 2024b; Tewel et al., 2024), LIGER additionally injects the previous step visual latent embedding into the frozen text-to-image diffusion model, generating coherent images for different steps. To further refine the object attribute in the images and avoid over-consistent, a MLLM receives multi-modal in-context prompting and tells the rectifying solutions. Various editing tools deal with errors including attribute error, object redundancy, identity inconsistency, and logic misunderstanding. Then the visual memory is calibrated to the embedding of the edited image via a latent inversion procedure, avoiding the error affecting future step image generation. Having this step-by-step generation manner, LIGER is capable of tasks with arbitrary steps without training.

To evaluate whether the generated visual instructions align with human comprehension, we curate a benchmark containing 569 long-horizon tasks along with human-annotated ground truth expressions and logic relations. Moreover, we evaluate the method from semantic alignment, logic correctness, and illustrativeness. Results show that LIGER surpasses baseline methods by a large margin. User studies and qualitative comparisons further verify that visual instructions generated by LIGER are more illustrative. In summary, the contribution of this paper includes:

(1) We propose LIGER, the first training-free framework generating visual instructions for long-horizon tasks.

(2) History prompts, visual memory, and self-reflection are introduced to promise logic coherent and object property accuracy. Inversion-based memory calibration is devised to avoid exposure bias.

(3) A dataset for long-horizon tasks with human-annotated expressions is curated to evaluate the effectiveness of LIGER.

## 2 RELATED WORK

### 2.1 IMAGE GENERATION AND EDITING

Recent advances in multi-modal diffusion models (Ramesh et al., 2022; Koh et al., 2024; Peebles & Xie, 2023; Saharia et al., 2022; Ho et al., 2020; Song et al., 2020) show a remarkable ability to generate images in high fidelity. Among these models, Latent diffusion models (LDMs) (Rombach et al., 2022) show strong robustness and semantic richness since the denoising process is conducted on the latent space. Based on LDMs, researchers further exploit exciting application topics including controllable image generation (Zhang et al., 2023; Mou et al., 2024b; Liang et al., 2024; Ma et al., 2024b), personalized generation (Ruiz et al., 2023; Kumari et al., 2023; Shi et al., 2024a; Gal et al., 2022), coherent generation Zhou et al. (2024b); Tewel et al. (2024), image editing (Brooks et al., 2023; Hertz et al., 2022; Nichol et al., 2021; Kim et al., 2022; Mou et al., 2023; Shi et al., 2024b; Mou et al., 2024a), etc. Storydiffusion (Zhou et al., 2024b) and Consistory (Tewel et al., 2024) share a similar idea of KV sharing to generate content-consistent images in a training-free manner.

Image editing, different from previous image generation tasks, involves manipulating the contents of the given image (Pan et al., 2023). There are various settings for editing, including text-driven (Tumanyan et al., 2023; Cao et al., 2023; Kawar et al., 2023), location-based (Chen et al., 2024b; Avrahami et al., 2023; Nichol et al., 2021), appearance modulation (Chen et al., 2024a; Mou et al., 2023), object moving (Pan et al., 2023; Mou et al., 2024a), etc. Common techniques for text-guided editing involve modifying the latent attention module *e.g.* MasaCtrl (Cao et al., 2023) or fine-tuning a model *e.g.* Instructpix2pix and SmartEdit (Brooks et al., 2023; Huang et al., 2024). Location-based editing leverages the region restriction prior like bounding box, mask, or even point (Ling et al., 2023). Our method utilizes different image editing methods to rectify the errors in the image.

### 2.2 TASK INSTRUCTION GENERATION

Generating procedures for a task is a popular research topic as it has potential application scenarios like intelligent assistants (Shen et al., 2024; Surís et al., 2023; Yang et al., 2024b), embodied agents navigation (Liu et al., 2023; 2024) and instruction comprehension (Xu et al., 2023), etc. This paper focuses on visual instruction generation, *i.e.* generating a series of images to explain a task. Previous work like TIP (Lu et al., 2023) and MGSL (Wang et al., 2022) generates textual instructions for the tasks based on the visual information. StackDiffusion (Menon et al., 2024) is the first method for generating coherent visual instructions, which is trained on step-wise annotated VSGI dataset (Yang et al., 2021). However, the step number for a task is restricted. GenHowTo (Damen et al., 2024) infers states before and after actions by learning from instructional videos. Sequential Latent Diffusion Model (Bordalo et al., 2024) trains a model to output coherent text prompts for the text-to-image diffusion model, therefore generating coherent images. Phung *et al.* Phung et al. (2024) propose a training-free method yet the utmost step length is 5. Different from previous methods, LIGER is a training-free method that can deal with long-horizon tasks having large step lengths.

### 2.3 TOOL-BASED METHODS

As the growing emergent capabilities of LLMs (Achiam et al., 2023), researchers deal with complex vision and natural language tasks (Yao et al., 2022; Ma et al., 2024a) by using surrogate tools (Schick et al., 2024) or programming languages, pioneer works include VisProg (Gupta & Kembhavi, 2023), ViperGPT (Surís et al., 2023), HuggingGPT (Shen et al., 2024), etc. In the image and video generation area, LLMs are widely used for arranging layouts (Gani et al., 2023; Lin et al., 2023; Lian et al., 2023; Yang et al., 2024a), enriching textual prompts (Cheng et al., 2024; Long et al., 2024; Yuan et al., 2024; Zhuang et al., 2024), tool calling (Wang et al., 2024), verification (Wu et al., 2024). Our method is also a tool-based framework unleashing the strong reasoning ability of Multi-modal Large Language Models (MLLMs) to call tools, enrich textual information, and do self-reflection.

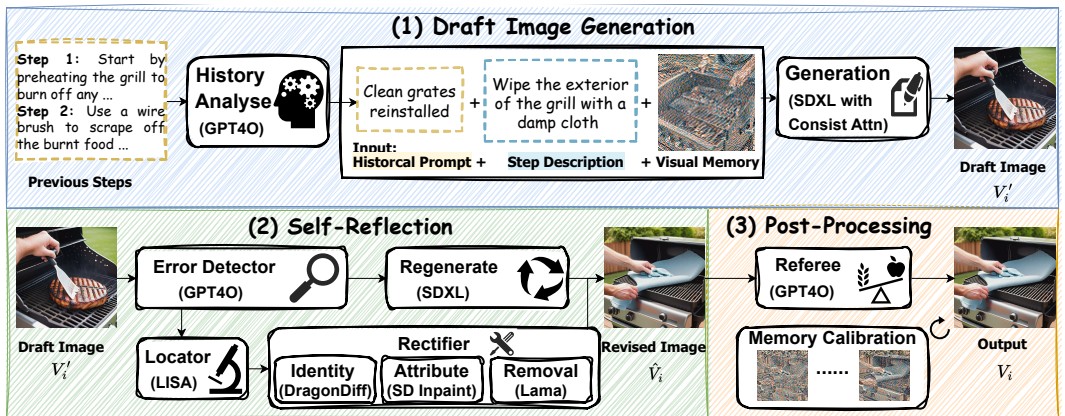

Figure 2: **Pipeline overview.** LIGER generates visual instructions step-by-step, starting with (1) generating a draft image taking the visual memory, step description and historical prompt as input. (2) The error detector identifies the error and the corresponding tool fixes it, generating a revised image. (3) The referee tool compares the two images and selects one as the final output. LIGER further uses inversion-guided visual memory calibration for future step generation.

## 3 METHOD

The overall pipeline of LIGER is shown in Figure 2. Harnessing the visual memory and historical prompt, LIGER generates a draft image for each step. Self-reflection mechanism corrects the errors in the draft images. To prevent error accumulation in the long horizon generation procedure, LIGER calibrates the visual memory according to the edited image through inversion.

### 3.1 HISTORY-AWARE DRAFT IMAGE GENERATION

Given a set of step descriptions $\mathbb{S}^n$ for a task $Q$ of $n$ steps, our goal is to generate a series of coherent images $\mathbb{V}$ for corresponding descriptions without training. To this end, a frozen text-to-image diffusion model generates a draft image $V_i'$ for step $i$ for each step in the task. The diffusion model generates a single image through iterative denoising steps. Specifically, a U-Net network $U$ predicts the noise

$$\epsilon_t = U(\boldsymbol{z}_t, \boldsymbol{c}), \tag{1}$$

where $\boldsymbol{z}_t$ is the latent representation at timestep $t$ and $\boldsymbol{c}$ is the textual condition. Naively generating individual images using the step description ignores the continuity between steps. Therefore, we first introduce the historical prompt and visual memory to enhance consistency.

**Historical prompt.** Each step description $S_i \in \mathbb{S}$ often describes an incremental action relative to the previous scene settings. For instance, in a task *cooking potato chips*, two consecutive steps are: *place the potato chips on a paper towel to drain excess oil* and *seasoning with salt and pepper*. Without context, the text-to-image diffusion model is unaware that salt and pepper should be added to the potato chips. Motivated by this, we use an LLM to generate a description $H_i$ for each step that specifies which objects from the previous steps should appear in the current step. The text condition $\boldsymbol{c}$ for the diffusion model is formulated as

$$\boldsymbol{c} = E_T(S_i, H_i), \tag{2}$$

where $E_T$ is the text encoder network.

**Visual memory sharing.** Merely using the historical prompt results in generating objects with varied appearances and backgrounds. To address this issue, inspired by StoryDiffusion (Zhou et al., 2024b), we incorporate visual embeddings from the previous step as the visual context. When generating the draft image $V_i'$ of step $i$, we randomly sample several visual feature tokens $p_{i-1} \in \mathbb{R}^{M \times C}$ of the previous image $V_{i-1} \in \mathbb{V}$ and inject them into the self-attention operation in the U-Net. Here $M$ represents the number of sampled tokens and $C$ is the number of feature channels. The query input of the attention operation is the current image feature tokens $p_i \in \mathbb{R}^{N \times C}$, the key and value inputs are the concatenation of $p_{i-1}$ and $p_i$. The procedure can be formulated as:

$$\begin{aligned} Q_i = W^q \boldsymbol{p}_i, K_i = W^k[\boldsymbol{p}_i, \boldsymbol{p}_{i-1}], V_i = W^v[\boldsymbol{p}_i, \boldsymbol{p}_{i-1}], \\ O_i = Attention(Q_i, K_i, V_i), \end{aligned} \tag{3}$$

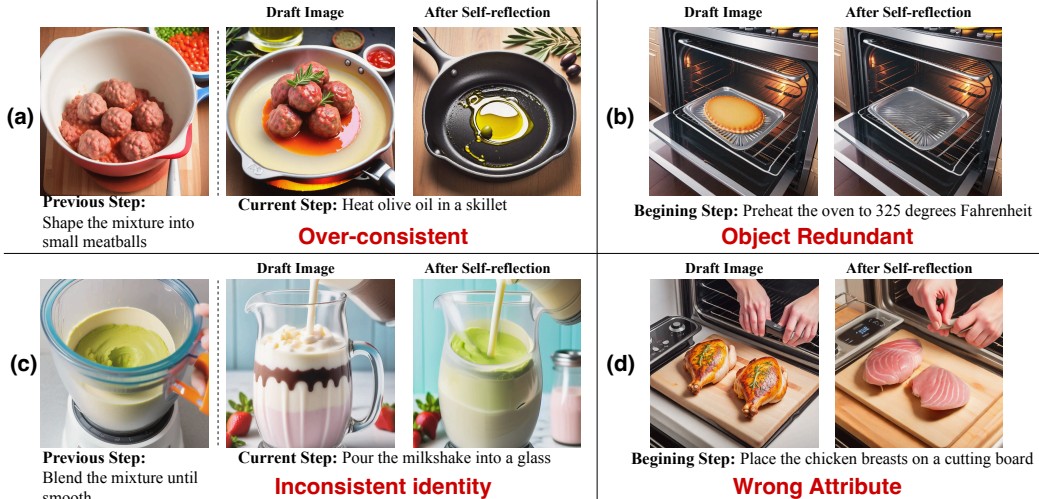

Figure 3: Visualization of different error types and the effect of self-reflection. The motivation of self-reflection is to rectify errors including (a) over-consistent, (b) object redundant, (c) inconsistent identity, and (d) wrong attributes.

where $W^q$, $W^k$, $W^v$ are the linear projection layers for the query, key, and value respectively. The output feature $O_i$ is used as the input of the next layer in the UNet $U$. Note that neither the historical prompt nor the visual memory are provided in the first step of any task.

## 3.2 TOOL-BASED SELF-REFLECTION

Empirically, we observe errors in the draft images as illustrated in Figure 3. Leveraging the advanced multi-modal capabilities of MLLMs, LIGER employs the state-of-the-art GPT4O model as an error detector to identify errors across four aspects, then output tool calling instructions to revise the draft images. For accuracy in error recognition, the error detector is prompted with multimodal in-context examples. The prompt template is attached in the appendix.

**Over-consistent.** In long-horizon tasks, not all steps necessarily require visual continuity. For example, consider the task of *cooking wanton noodles* where the steps *Drain the noodles and rinse with cold water* and *In a separate pan, heat some oil* are sequential yet independent. The former step concludes noodle preparation, while the latter step initiates cooking with different ingredients. These steps lack logistic connection, making consistency between the two images unnecessary. Breaking this consistency can help users recognize the transition to a new step. To address the over-consistent issue, the error detector assesses whether to maintain or disrupt the continuity. If breaking consistency is required, the error detector outputs the error rectification instruction in the format of $Regenerate(New\ text)$, then regenerates an image according to the new description.

**Identity inconsistent.** Despite historical prompt and visual memory contributing to global visual consistency, local details occasionally remain misaligned, as depicted in Figure 3. To enhance local consistency, LIGER employs an intuitive method that aligns object appearances across images. Specifically, the error detector compares objects in successive images, identifying whether two objects should have similar appearance with the command $Modify(object\ in\ V_i', object\ in\ V_{i-1})$. Subsequently, a locator tool, *i.e.* LISA (Lai et al., 2024) outputs the masks of the objects according to the object descriptions generated by the error detector. Then the identity-keeping tool *i.e.* DragonDiffusion (Mou et al., 2023) receives the masks and modifies the object appearance in the current image to match the previous image.

**Wrong attribute.** Correct object attributes such as color, shape, and state are crucial for instructions. For instance, considering the tasks of *baking chicken wings*, the model may incorrectly generate cooked chicken wings at the *seasoning the prepared chicken wings* step, where they should be raw. To address this problem, the error detector describes the desired attributes for an object with the instruction $Add(new\ description, object\ in\ V_i')$. The same locator tool segments the object, then an attribute reformulation tool *i.e.* SD inpainting Rombach et al. (2022) generates an image with modified object attributes according to the object mask.

**Redundant object.** The last type of error is object hallucination, where frozen text-to-image

diffusion models sometimes generate irrelevant objects for a step description. For instance, in Figure 3 (b), the image illustrating *preheating the oven* mistakenly includes bread in the pan. The error detector flags the object to be removed in a format of $Remove(object\ in\ V_i')$, and the locator tool pinpoints the specific region. LIGER opts for the widely used LAMA (Suvorov et al., 2022) as an object removal tool. The tool removes the corresponding part of the image given the object mask. LIGER evaluates the image across these four aspects iteratively and only modifies the draft image for once. In other words, once an error is detected, the verification procedure halts, and the corresponding editing operation is applied to the draft image. It is also worth noting that the over-consistent and identity inconsistent errors are verified based on two consecutive steps, while wrong attribute and redundant object are conducted as single-image verifications. The execution order of the pipeline is detailed in Algorithm 1. Consequently, for the draft image of the first step in each task, LIGER only performs attribute modification or object removal. Having the various tools collaboratively verify the images, LIGER generates illustrative visual instructions for long-horizon tasks with accurate logic in a self-reflection manner.

---

**Algorithm 1** Single Step Self-reflection

---

**Input:** Draft Image $V_i'$, Previous Image $V_{i-1}$, Step Description $S_i, S_{i-1}$, and Task $Q$.
**if** $i = 0$ **then**
  $\quad \mathbb{A} \leftarrow [\textit{Attribute, Object}]$
**else**
  $\quad \mathbb{A} \leftarrow [\textit{Relation, Identity, Attribute, Object}]$
**end**
**for** $A$ *in* $\mathbb{A}$ **do**
  $\quad$**if** $A$ *in* [*Attribute, Object*] **then**
    $\quad\quad error \leftarrow \text{Detect}(V_i', S_i, Q)$
  $\quad$**else**
    $\quad\quad error \leftarrow \text{Detect}(V_i', S_i, Q, S_{i-1}, V_{i-1})$
  $\quad$**end**
  $\quad$**if** *error is detected* **then**
    $\quad\quad \hat{V}_i \leftarrow \text{Rectify}(V_i', S_i, Q)$
    $\quad\quad V_i \leftarrow \text{Compare}(V_i', \hat{V}_i)$
    $\quad\quad$**break**
  $\quad$**end**
**end**
**if** $V_i = \hat{V}_i$ **then** Refresh($\hat{V}_i$) **end**
**Output:** Final Image $V_i$,

---

## 3.3 Judgement and Memory Calibration

The aforementioned tool-based self-reflection generates a revised image $\hat{V}_i$. Yet every rose has its thorn, self-reflection sometimes produces low-quality images or makes incorrect judgments during editing. To stabilize the pipeline predictions and improve robustness, we devise a referee tool to compare the draft image with the revised image. The referee evaluates both the quality and semantic alignment of the images and selects the better one as the final result $V_i$. For more details, refer to the prompt template provided in the appendix. Since LIGER generates images step by step, with visual memory providing visual continuity between steps, any error in the output image $V_i$ impacts the memory and can accumulate in subsequent steps of image generation. To prevent this exposure bias, we propose inversion-guided visual memory calibration to update the memory.

**Inversion-guided visual memory calibration.** As discussed in Section 3.1, the visual memory is a set of image feature tokens sampled from the previous generation step $p_{i-1} \in \mathbb{R}^{M \times C}$. These tokens are saved during the denoising process of the draft image, which exhibits a discrepancy with the features of the revised images. Since the revised image is generated in a post-processing manner, storing the feature tokens alongside the generation process is inapplicable. However, the sampling process can be reversed using DDIM inversion which is formulated as:

$$\boldsymbol{x}^{t+1} = \sqrt{\alpha_{t+1}/\alpha_t} \cdot \boldsymbol{x}^t + \sqrt{\alpha_{t+1}}\left(\beta_{t+1} - \beta_t\right) \cdot \epsilon_t, \tag{4}$$

where $\alpha_t$ is the variance schedule depend on timestep $t$, and the step-wise coefficient is set to $\beta_t = \sqrt{1/\alpha_t - 1}$. $\epsilon_t$ is the noise predicted by the U-Net according to Eq 1. This allows us to obtain the attention output of the U-Net during the inversion procedure. Therefore, for the revised images, we apply this inversion operation over the same number of timesteps as in the generation procedure, effectively calibrating the visual memories to current image $V_i$ features. Correcting the visual memories prevents accumulated errors affecting subsequent image generation procedures.

## 4 Experiments

### 4.1 Implementation Details

For the historical textual prompt, the error detector and referee, we use GPT-4O (Achiam et al., 2023) introduced by OpenAI. The draft image generation uses the SDXL (Podell et al., 2023) with

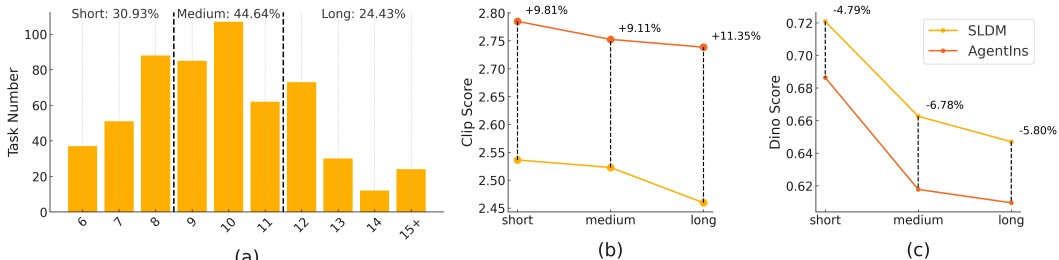

Figure 4: Dataset statistics and the influence of the step length of tasks.

a guidance scale of 5 along with the Free-U plugin (Si et al., 2024). The DDIM generation and inversion timesteps are set to 50. In terms of the visual memory, we set the number of the previous step image feature token $M$ to half of the sequence length $N$, in other words, $M = N/2$. For the location tool, we leverage the LISA-7B model (Lai et al., 2024) to balance the performance and computing resources requirement. All experiments are conducted on a single RTX A6000 GPU.

## 4.2 DATASET

Effective visual instructions for long-horizon tasks should help users quickly understand complex procedures, but evaluating this capability remains challenging. Existing datasets lack appropriate evaluation methods for this aspect. To address this gap, we curate a new textual dataset consisting of 569 long-horizon tasks. These tasks are extracted from different resources including Howto100M (Miech et al., 2019), Youcook2 (Zhou et al., 2018), and RecipeQA (Yagcioglu et al., 2018). The tasks focus on the recipe domain, as cooking procedures typically involve strong logical relations between steps and require multiple stages. Specifically, we prompt the GPT4O model with in-context samples to filter out tasks that are hard to illustrate and tasks that are easy to accomplish, *e.g. How to prepare a family meal for 20 people*. The LLM then outputs step-by-step action descriptions for each task. Unlike existing planning datasets (Menon et al., 2024; Lu et al., 2023), our dataset offers following novel features:

**Long-horizon tasks.** The average number of steps per task is 9.8, with a minimum of 6 steps and a maximum of 17. The detailed distribution is shown in Figure 4 (a). We categorize the tasks into three types: short (6-8 steps), medium (9-11 steps), and long (12 or more steps).

**Manual annotations for step logics.** For each task, we ask human annotators to select a pair of consecutive steps with continuous logic and another pair with logically independent steps. Our intuition is that the images corresponding to logically consistent steps should exhibit visual continuity, while the images of locally independent steps should be visually distinct.

**Human-written ground truth descriptions reflecting comprehension.** We introduce a novel annotation for evaluating illustrative images. Since step descriptions often omit details about object attributes, we ask the annotators to write a sentence describing what components should appear in the illustrative image for every step. These sentences reflect the appearance and state of the objects with previous steps information. For example, the step *Arrange the chicken wings on the wire rack* from task *How to bake chicken wings*, one can infer the wings are raw and ready for baking. Therefore, a suitable illustrative expression could be *The raw chicken wings are neatly arranged in a single layer on the wire rack, with the spices and oil giving the skin a glossy, seasoned appearance.* These expressions allow us to evaluate whether the generated images match human expectations of how an illustrative image should look. Annotation examples are provided in the appendix.

## 4.3 BASELINES

To thoroughly evaluate the effectiveness of LIGER and its components, we conduct both quantitative and qualitative comparisons with different baselines including: (1) **Frozen SDXL (Podell et al., 2023).** We simply generate visual instructions for the tasks using a frozen SDXL model prompted with the vanilla textual step descriptions. (2) **Frozen SDXL + Visual memory (+V).** The image generation model is provided with the visual memory while the text prompts remain vanilla step descriptions. (3) **Frozen SDXL + Historical prompt (+H).** The text prompt for the frozen SDXL model is modified by concatenating the step description and the historical prompt. No visual memories are provided. (4) **Frozen SDXL + Visual Memory + Historical prompt (+V+H).** The image generation model is equipped with both visual memory and the historical prompt. This baseline can

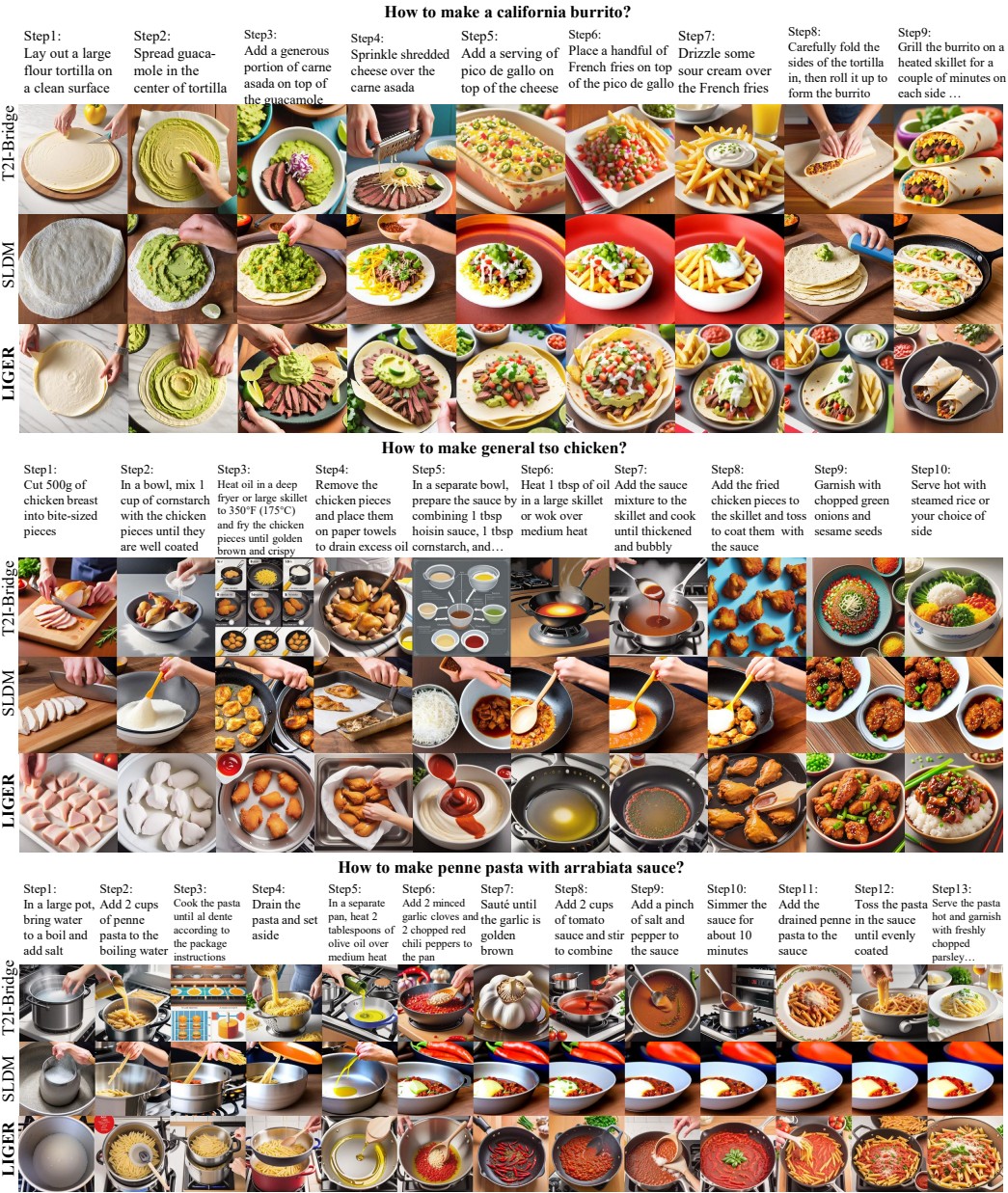

Figure 5: Detailed qualitative comparisons on different long-horizon tasks. Zoom in to see details.

also be considered LIGER without self-reflection. (5) **T2I-Bridge** (Lu et al., 2023) uses an LLM to imagine what the image for each step should depict based on the step descriptions. T2I-Bridge represents a type of re-captioning method. (6) **Sequential Latent Diffusion Model (SLDM)** (Bordalo et al., 2024) trains a language model to produce coherent captions for the steps of a task and uses a sequential context decoder to establish visual connections between images. Note that the text-to-image generation diffusion model is still frozen in SLDM.

## 4.4 QUANTITATIVE EVALUATION

To assess the effectiveness of LIGER, we conduct a detailed quantitative comparison including:
**Automatic evaluation.** We calculate several metrics using pre-trained models. First, we evaluate the semantic alignment between the images and human-annotated ground truth expressions by calculating the CLIP (Radford et al., 2021) similarity. These curated expressions reflect human understanding of each step. Hence a higher CLIP-Score indicates that the images are more relevant to the expressions, implying that the images are more illustrative for human comprehension.

| Method | Automatic evaluation | | | GPT evaluation | | |
|---|---|---|---|---|---|---|
| | CLIP-Score↑ | DINO-Score ↓ | BERT-Score ↑ | Semantic↑ | Logic↑ | Illustrative↑ |
| T2I-Bridge | 2.4350 | 0.8576 | 0.8669 | 3.4717 | 2.5843 | 2.5150 |
| SLDM | 2.5054 | 0.6746 | 0.8694 | 3.3634 | 2.7286 | 2.5771 |
| **Ours** | **2.7555** | **0.6338** | **0.8743** | **4.1141** | **3.0595** | **3.0536** |

Table 1: Automatic quantitative evaluation and GPT evaluation results.

The second metric tests the logic correctness between consecutive steps. To evaluate image similarity, we use the DINO-v2 (Caron et al., 2021; Oquab et al., 2023) model and calculate the average $l_2$ Distance between the embeddings of the two images for the annotated step pairs. Inspired by the Signal-to-Noise Ratio formulation, we define the DINO-Score $D_s$ as the $l_2$ distance between coherent steps divided by the $l_2$ distance between independent steps which can be expressed as $D_s = l_2^p/l_2^n$. This metric evaluates the ability to generate consistent images for logically coherent steps and distinct images for unrelated steps. A lower DINO-Score indicates higher logical accuracy.

The last metric evaluates the method performance in a modality-transfer test. Our intuition is that illustrative visual instruction should help people summarize or describe the steps in text. Therefore, we transfer the images back into text and measure the textual similarity with the annotated descriptions. Specifically, we adopt the widely-used BLIP-2 (Li et al., 2023) model to generate captions for images, then calculate the BERT-Score (Zhang et al., 2019) between the captions and descriptions. A higher BERT-Score represents the image is more illustrative. The results shown in Table 1 demonstrate that LIGER significantly outperforms the baseline methods.

**GPT evaluation.** We further harness the advanced logical reasoning and multi-modal perception ability of MLLMs to evaluate the methods. Specifically, we prompt the GPT4O model to rate how well each individual image aligns with its corresponding description. Then we input the entire image series to the MLLM and ask it to rate whether the image series is illustrative with correct logics. The rating ranges from 1 to 5, where 1 represents low quality and 5 indicates perfect quality. The results are shown in Table 1, and the prompt templates are attached in the appendix.

**User study.** We invite 20 participants for the user study, with each person asked to select the best generation results for 15 tasks. Participants rate aspects

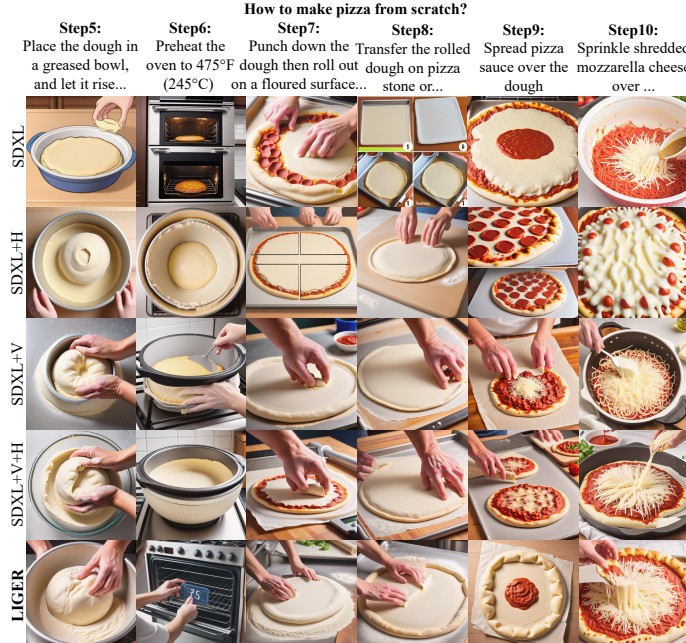

Figure 6: Qualitative ablation on different components.

including semantic alignment, logical correctness, and task illustration. Results in Table 3 show that LIGER generates visual instructions that better match user preferences while maintaining semantic alignment and logic accuracy.

## 4.5 QUALITATIVE COMPARISONS

The overall qualitative comparisons between LIGER and baseline methods are shown in Figure 5. We provide a detailed comparison of LIGER with two prior works, namely T2I-Bridge and SLDM. For the task *How to make a California burrito*, both T2I-Bridge and SLDM overlook that the seasoning and ingredients are added to the tortilla in Steps 3 to 7. In contrast, LIGER clearly illustrates the progressive process of adding different ingredients. Additionally, LIGER correctly visualizes the burrito being wrapped and heated in a skillet. For the task *How to make general*

*tso chicken*, LIGER presents a smooth sequence, showing the process of frying the chicken pieces, making the sauce, combining sauce with chicken, and serving with rice. In comparison, SLDM omits the chicken pieces in Step 2 and incorrectly shows the finished dish in Step 5. T2I-Bridge lacks visual continuity, making it hard to comprehend. To further demonstrate the effectiveness of LIGER in long-horizon tasks, we visualize the results for the task *How to make penne pasta with arrabbiata sauce* consisting of 13 steps. SLDM shows an over-consistent process during cooking, while T2I-Bridge generates distinct images. In contrast, LIGER accurately illustrates the procedure.

## 4.6 ABLATION STUDY

**Effectiveness of different components.** We provide both qualitative and quantitative comparisons in Figure 6 and Table 2. Results show that adding historical prompts and visual memory both improve the alignment between image and text semantics while also increasing logical accuracy. Additionally, these two components complement each other. When self-reflection is introduced, we observe a performance gain of +0.04 in CLIP-Score, a reduction of -0.112 in DINO-Score, and an improvement of +0.002 in BERT-Score, demonstrating the importance of self-reflection. In Figure 6, we observe that self-reflection correctly identifies which steps should be visually coherent and which steps should be distinct. Moreover, LIGER effectively shows the process of transforming pizza dough into a raw pizza. Essentially, the historical prompt and visual memory enhance visual continuity, while self-reflection aligns the images with human comprehension.

We further provide an example to highlight the importance of visual memory calibration in Figure 7. For Step of *season the steak*, the steak should be raw, yet the draft image incorrectly shows a cooked appearance. After correcting the attribute, the subsequent step should also depict the steak as raw since the description does not indicate a state change. Without memory calibration, the steak in the next step still appears cooked, but with calibration, the steak is correctly shown in a raw state.

| | CLIP-Score ↑ | DINO-Score ↓ | BERT-Score ↑ |
|---|---|---|---|
| SDXL | 2.5837 | 0.8516 | 0.8699 |
| SDXL+V | 2.6251 | 0.8239 | 0.8719 |
| SDXL+H | 2.6842 | 0.8224 | 0.8707 |
| SDXL+V+H | 2.7168 | 0.7459 | 0.8721 |
| **Ours** | **2.7555** | **0.6338** | **0.8743** |

Table 2: Ablation on different components of LIGER.

| Method | Semantic | Logic | Illustrative |
|---|---|---|---|
| T2I-Bridge | 24% | 18.3% | 22.3% |
| SLDM | 11.7% | 21% | 9.3% |
| **Ours** | **64.3%** | **60.7%** | **68.3%** |

Table 3: User study on image-text semantic matching, logic continuity and illustrative.

Figure 7: Example of visual memory calibration.

**Influence of task step length.** In Figure 4 (b) and (c), we present the CLIP-Score and DINO-Score for tasks of varying lengths, comparing LIGER with SLDM. As the number of task steps increases, the CLIP-Score of SLDM decreases significantly, while LIGER maintains stable performance. Additionally, the relative improvement in DINO-Score increases for medium and long tasks, indicating LIGER is robust to long-horizon tasks.

## 5 CONCLUSION

In this paper, we propose LIGER, the first training-free framework for long-horizon visual instruction generation. LIGER first leverages historical prompts and visual memory to generate draft images step-by-step, enhancing continuity between images in long-horizon tasks. The tool-based self-reflection rectifies four types of errors in the draft images including over-consistent, identity inconsistent, wrong attributes, and object redundant. LIGER also deploys inversion-guided visual memory calibration to prevent error accumulation in the sequential image generation procedure. We also curate a new benchmark testing the alignment of generation results with human comprehension. We hope this work inspires future research on instruction generation.

## 6 ACKNOWLEDGEMENTS

This work is supported by the National Natural Science Foundation of China (U2336212). This work is also supported in part by "Pioneer" and "Leading Goose" R&D Program of Zhejiang (No.2024C01142). We are grateful for the user study participants. This work was partially supported by ZJU Kunpeng&Ascend Center of Excellence. We also thank Dr Xiao Pan for discussing about the paper writing.

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

## A  APPENDIX

### A.1  ADDITIONAL RESULTS

Additional qualitative results are shown in Figure 8 and Figure 9.

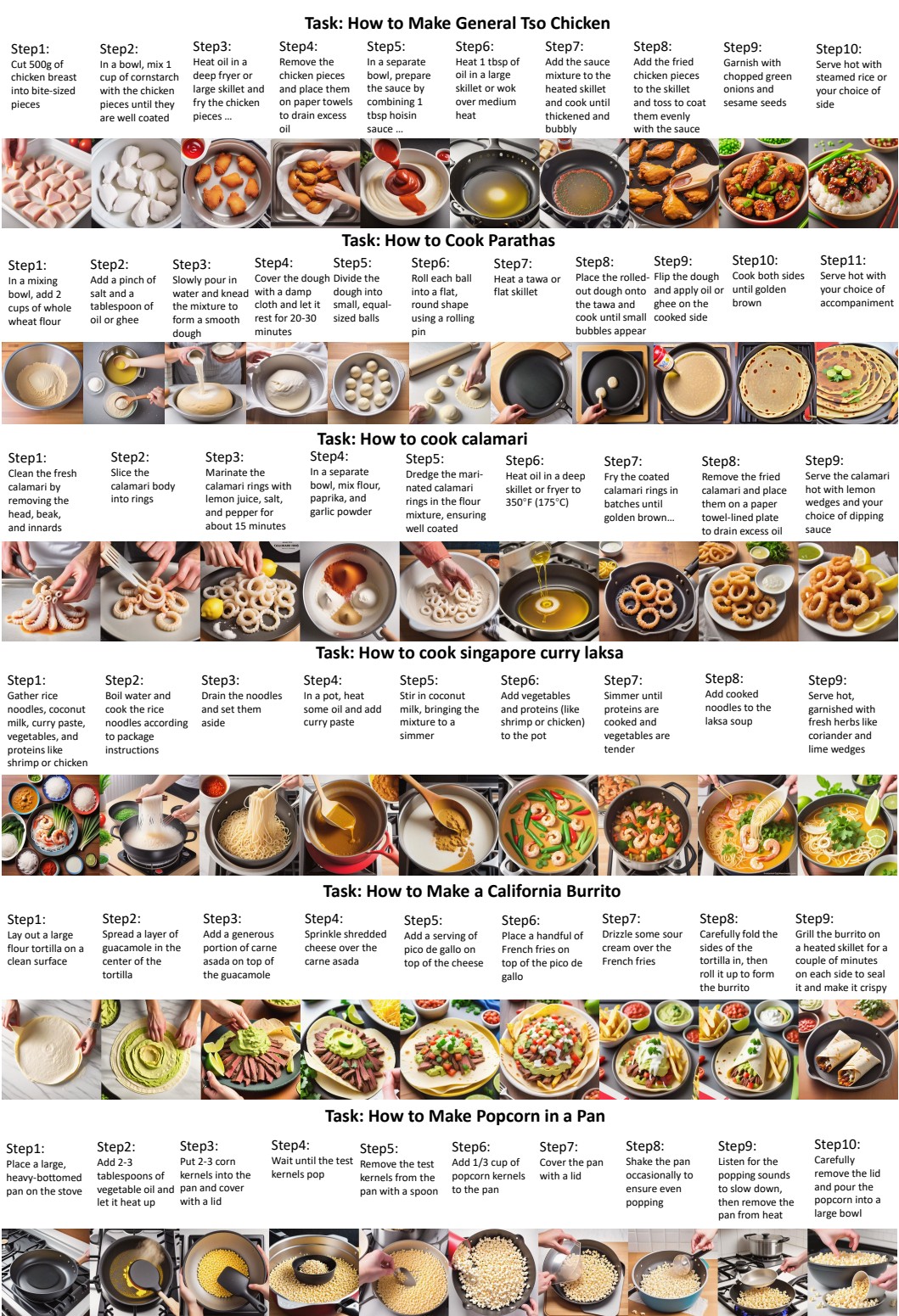

Figure 8: Additional qualitative results generated by LIGER. Zoom in to see the detail.

## A.2 Additional Quantitative Experiments

We present the following quantitative ablation studies in addition:

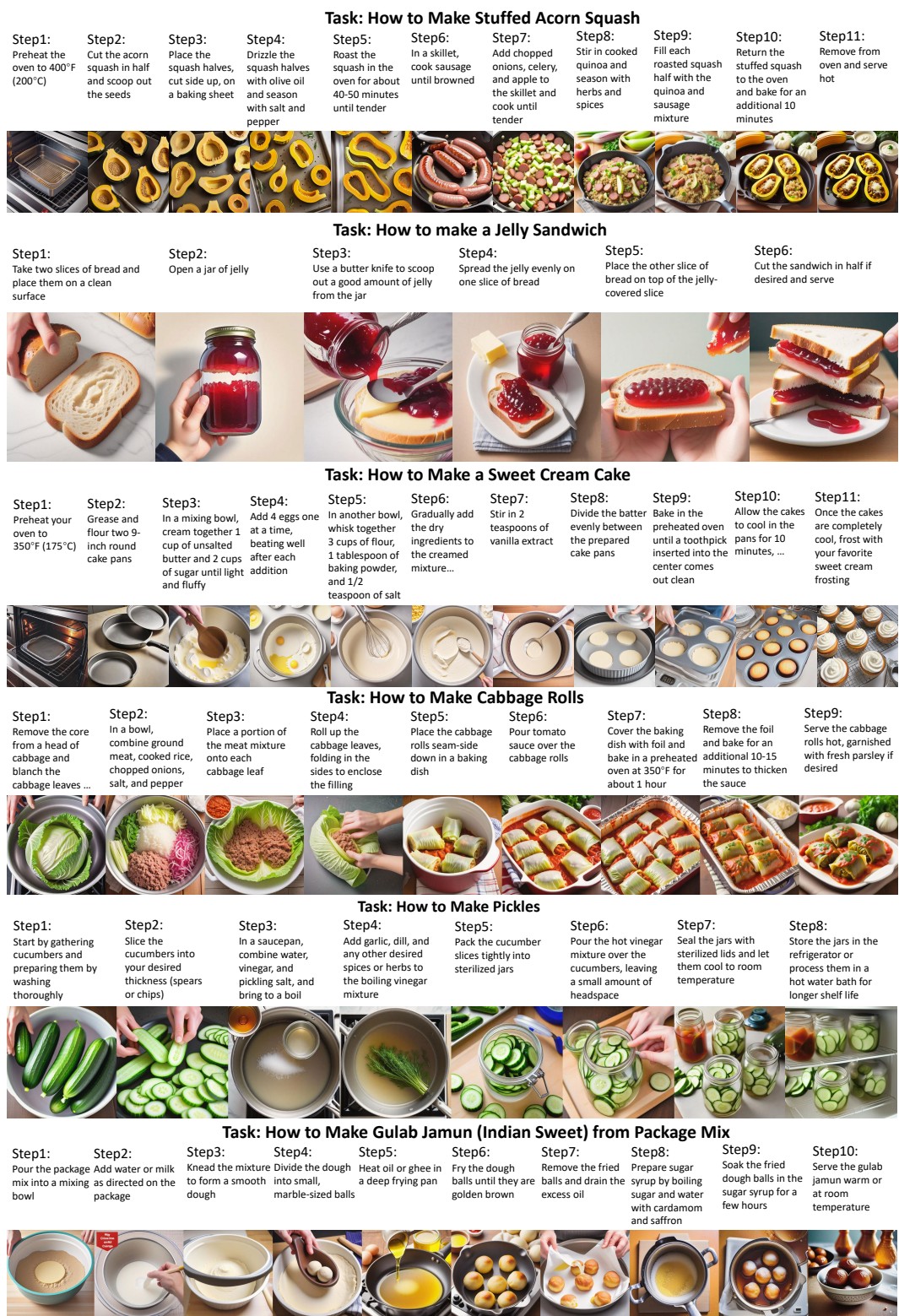

Figure 9: Additional qualitative results generated by LIGER. Zoom in to see the detail.

(1) Effectiveness of different components. To thoroughly evaluate the importance, we conduct further ablation and report results in Table 4.

| Method | CLIP-Score↑ | DINO-Score ↓ | BERT-Score ↑ |
|---|---|---|---|
| SDXL | 2.5837 | 0.8516 | 0.8699 |
| SDXL+V | 2.6251 | 0.8239 | 0.8719 |
| SDXL+H | 2.6842 | 0.8224 | 0.8707 |
| SDXL+R | 2.7270 | 0.7346 | 0.8732 |
| SDXL+H+V | 2.7168 | 0.7459 | 0.8721 |
| SDXL+V+R | 2.7428 | 0.7053 | 0.8734 |
| SDXL+H+R | 2.7440 | 0.6653 | 0.8740 |
| LIGER | 2.7555 | 0.6338 | 0.8743 |

Table 4: Different combination of Self-reflection and other components.

(2) Robustness towards MLLMs. LIGER integrates the strong GPT4o as the error detector and referee agent. To evaluate the influence of MLLM, we conduct an ablation by substituting the GPT4o model with two open-source models *i.e.* Pixtral-12B and QwenVL-7B. Automatic metric comparison is shown in Table 5. There is a performance drop when using open-source models. We empirically find the output of the open-source model lacks reasoning ability and detail region comprehension ability, leading to misunderstanding the error type or missing the obvious errors.

| Method | CLIP-Score↑ | DINO-Score ↓ | BERT-Score ↑ |
|---|---|---|---|
| SDXL | 2.5837 | 0.8516 | 0.8699 |
| LIGER(QwenVL-7b) | 2.7244 | 0.7305 | 0.8725 |
| LIGER(Pixtral-12b) | 2.7316 | 0.7061 | 0.8716 |
| LIGER(GPT4o) | 2.7555 | 0.6338 | 0.8743 |

Table 5: Ablation on MLLMs.

(3) Variance test. We run another trial on the whole 569 tasks and report the result in Table 6. Results indicate that there is a relatively small variance in the three evaluation metrics. Never the less, LIGER still consistently outperforms baseline methods.

| Method | CLIP-Score ↑ | DINO-Score ↓ | BERT-Score ↑ |
|---|---|---|---|
| T2I-Bridge | 2.4350 | 0.8576 | 0.8669 |
| SLDM | 2.5054 | 0.6746 | 0.8694 |
| LIGER | 2.7555 | 0.6338 | 0.8743 |
| LIGER (new trial) | 2.7738 | 0.6276 | 0.8745 |

Table 6: Variance test results on the whole dataset.

(4) Image quality evaluation. We evaluate the image quality using the GPT4O model to rate the quality of individual images of the whole 569 tasks from 1 to 5 where a higher rating indicates higher quality. We further conduct a user study, 5 participants view 50 images generated by each method and picked the best image from the three methods, and the win rate is reported in Table 3.

| Method | GPT-score ↑ | User Win Rate ↑ |
|---|---|---|
| T2I-Bridge | 3.8525 | 41.2% |
| SLDM | 3.7078 | 11.6% |
| LIGER | 3.8976 | 47.2% |

Table 7: Image quality evaluation.

## A.3 Additional Qualitative Experiments

We present the following qualitative comparison in addition:

(1) Comparison with relative works. Figure 10 shows a comparison between LIGER, Consistory and StoryDiffusion. LIGER shows a clear object attribute change along the task procedure. Not that we also need to manually define a subject concept for Consistory and StoryDiffusion, which id not required by LIGER.

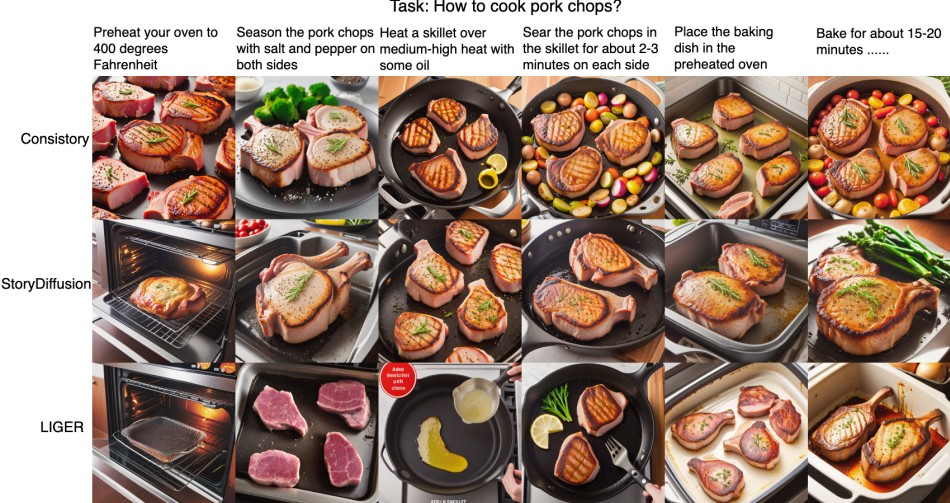

Figure 10: Additional qualitative results generated by LIGER. Zoom in to see the detail.

(2) Error analysis. Figure 11 shows two types of errors in LIGER. For the reasoning error case, the precious step is putting an egg in the batter and whisk. The current step is adding vanilla extraction. However, the error detector mistakenly believes the egg should be visible in the current step, which should not be after whisking. The referee agent finds the error and keeps the draft image as the final output. The right lane shows a case of generation error due to the location tool failure. The balls are mistakenly removed and the referee agent finds the mistake considering image quality and picks the draft image as the final output.

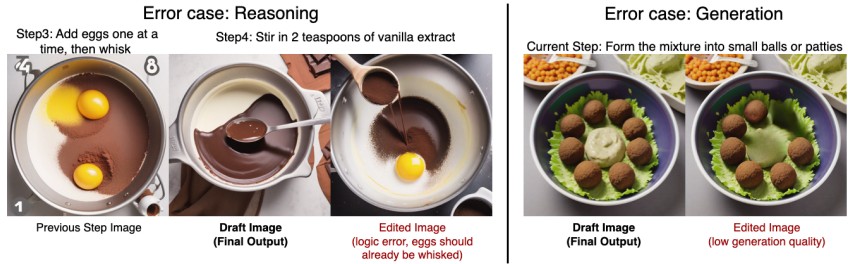

Figure 11: Error case analysis.

(3) Qualitative results in different scenarios. In Figure 12, we show the generalization ability of LIGER on tasks in other scenarios.

(4) Comparison on other short datasets. Figure 13 shows comparison on the recipeplan dataset with short abstract textual instructions. LIGER generates images fluently indicating the task procedure.

## A.4 LIMITATIONS

LIGER shows a strong ability to generate visual instructions for various tasks, yet there are still limitations. First is that the action generation is still uncontrollable. Future work may efficiently fine-tune the generation model to add illustrative actions in the images. Second, the amount of ingredients is not controllable. It is challenging for the frozen text-to-image diffusion models to identify how to visualize *1/2 cup of water and 1/4 teaspoon of salt*. We believe future research on generating videos for instructions might be an ideal way to show these details.

Another limitation is that since LIGER is a training-free framework, the generation quality depends on the pre-trained diffusion model. We find the current models struggle to generate fine-grained actions, or part of a complex structures. We believe LIGER can benefit from storng models.

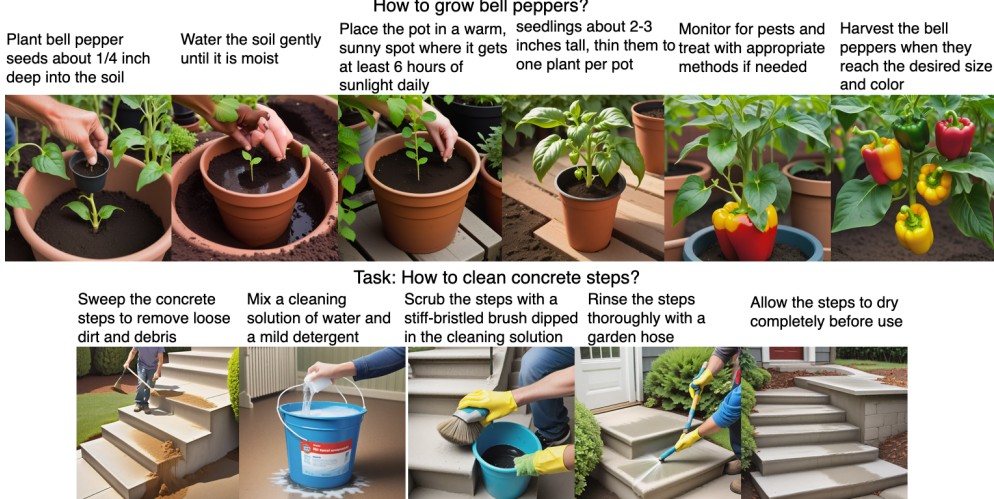

Figure 12: Qualitative results on other scenarios.

LIGER adopts many tools to collaboratively generate illustrative instructions, therefore the inference time is longer. A speed test over 50 randomly selected tasks using a locally deployed multi-modal large language model. LIGER takes around 120 seconds to generate instructions for a 10-step task while the frozen stable diffusion model takes around 60 seconds on a single A100 GPU. In the future, using quantilized models or conducting accelerating strategies may increase the efficiency.

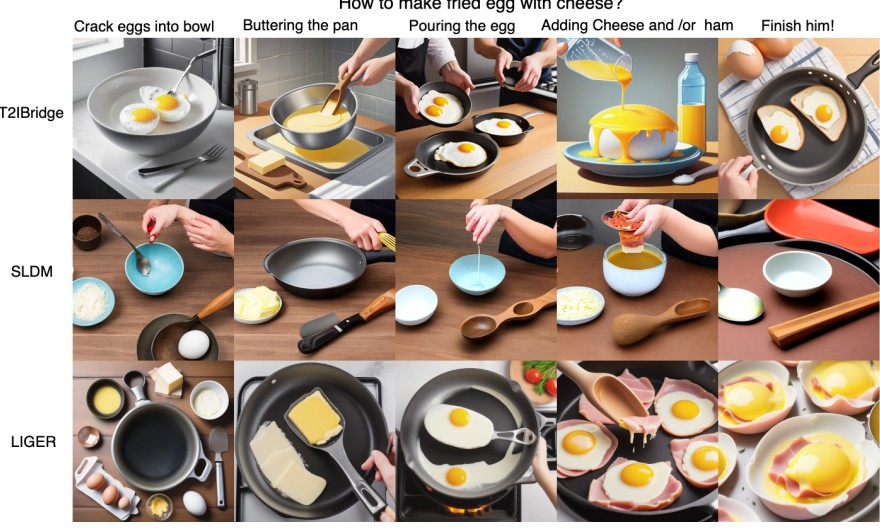

Figure 13: Qualitative results on recipeplan dataset.

## A.5 PROMPT TEMPLATES

We provide several detailed prompt templates here. The prompt for the four types of error are:

(1) Over-consistent:

```
Now I wish you to use logic reasoning ability to judge whether
the current image should be regenerated. If the previous and
current step action does not have logic correlation, the
previous scene description of current step description is wrong,
then the current image needs to be regenerated. You need to
change the step description and tell me in a format of:
*-Regenerate(new step description)-*. The new description should
describe in detail about what objects should be in the new image.
For example, {In-context example with reason}
```

```
If the logic between the two step action descriptions are
coherent, you just answer Correct, no error.
Please ignore the objects in the background and be tolerant to
errors that are not obvious. You must tell why to make the
choice and only correct the most obvious error with only one
operation. Now, for a procedure of {task}, the previous step
is {pre_step}, and the previous image is: {pre_image} the
current step description is {cur_step} and the current image
is: {cur_image}
```

(2) Identity:

```
Use logic reasoning ability to judge whether the subject object
should look exactly the same between the two images based on the
previous and current step description. If there is an object
should look totally the same but not, tell in a format of:
*-Modify(object in current image, object in previous image)-*.
For example, {In-context example with reason}. If the image is
correct, you just answer Correct, no error. You only consider
whether the foreground object appearance texture (not including
shape and size) should be the exactly the same. Please ignore
the objects in the background and be tolerant to errors that
are not obvious. You must tell why to make the choice and only
correct the most obvious error with only one operation.
Now, for a procedure of {task}, the previous step is {pre_step},
and the previous image is {pre_image}. The current step
description is {cur_step} and the current image is {cur_image}.
```

(3) Attribute:

```
We are generating illustrations for a procedure. Please evaluate
the image quality according to the current step description. You
need to identify whether the salient main object attribute
matches the step description. If the attribute is not ideal, you
need to tell me how to add it in a format of:
*-Add(object description, place to add the object)-*.
Must start with *-Add( and end with )-*. \
For example, {In-context example with reason}
If the image is correct, you just answer Correct, no error. You
only consider the forground salient object of the image. Please
ignore the objects in the background and be tolerant to errors
that are not obvious. You must tell why to make the choice and
only correct the most obvious error with only one operation.
The current procedure is {task} and the current step is
{cur_step}, and the image is {cur_image}
```

(4) Redundant:

```
We are generating illustrations for a procedure. Please
evaluate the image quality according to the current step
description. You need to identify whether there are
redundant objects. If redundant object exists, you need
to tell me in a format like:
*-Remove(object description)-*. Must start with *-Remove(
and end with )-*. For example, {In-context example with
reason}. The objects described in the Previous scene part
of the step description should not be regarded as
redundant object. If the image is correct, you just
answer Correct, no error. You only consider whether there
is obvious redundant object in foreground of the image.
```

```
Please ignore the objects in the background and be
tolerant to errors that are not obvious. You must tell
why to make the choice and only correct the most obvious
error with only one operation. The current procedure is
{task} and the current step is {cur_step}, and the image
is {cur_image}.
```

The prompt for comparing the draft image and the revised image is:

```
Please pick the better image between the two images
considering the image quality and the alignment with the
current step description. You only answer A or B within
one word. For example, {In-context example with reason}.
Now, consider the following step, {input_cur}, and the
image A is: {image_initial}, the image B is: {image_final}.
```

The prompt for GPT evaluation is:

(1) Single image evaluation

```
Rate the image from 1 (worst) to 5 (perfect) considering:
A. Does the image contains the objects should appear for
the text description?
B. The image does not contain unrelated objects?
C. According to the text description, imagine the subject
object attribute (adjective, state, color, texture), and
does the image show correct attributes?
Give a rate from 1 to 5 on each aspect within 30 words
in a format like A:rating*.
The text description is {input_overall} and the image is:
```

(2)Image series evaluation

```
Please rate the series images from 1 (worst) to 5 (ok)
considering:
A. In some consecutive steps, the images are coherent.
B. The image is diverse when the text descriptions deviate.
C. Overall, can the whole image series roughly describe
the coarse idea of the task?
Give a rate from 1 to 5 on each aspect. Do not be too
strict since the task is hard. The response should be in
a format of A:(number of ratings)* Reason: (reasons).
Considering the task of {task}. The text description for
each step is {steps} and the image series are:
```

A.6   DATASET EXAMPLES AND DISCUSSIONS

We showcase an example of *How to cook salmon fillet* in the annotated dataset:

```
Most unrelated step: 1 and 2
related step: 4 and 5 *
Step 1:
Action: Preheat your oven to 400°F (200°C).
Ground Truth Description: A modern metal oven is slightly
open with the display showing 400°F. The interior oven light
softly illuminates the empty metal racks inside, indicating
the oven is warming up. *
Step 2:
Ground Truth Description: A baking sheet is lined with
aluminum foil or parchment paper.
```

```
Action: Line a baking sheet with aluminum foil or
parchment paper. *
Step 3:
Ground Truth Description: The salmon fillet is placed
on the prepared baking sheet, skin side down, with the
pink flesh exposed for seasoning.
Action: Place the salmon fillet on the baking sheet,
skin side down. *
Step 4:
Ground Truth Description: Olive oil is being drizzled
over the top of the salmon fillet, giving it a glossy
sheen and helping to lock in moisture while baking.
Action: Drizzle olive oil over the salmon fillet. *
Step 5:
Ground Truth Description: The salmon is seasoned with
salt and pepper, and herbs or spices are sprinkled
over the top for added flavor.
Action: Season with salt and pepper, and add any
desired herbs or spices. *
Step 6:
Ground Truth Description: The baking sheet with the
seasoned salmon fillet is placed in the preheated oven.
Action: Place the baking sheet in the preheated oven. *
Step 7:
Ground Truth Description: The salmon is baking in the
oven for 12-15 minutes, turning opaque and flaking
easily with a fork when fully cooked.
Action: Bake for 12-15 minutes, or until the salmon is
cooked through and flakes easily with a fork. *
Step 8:
Ground Truth Description: The baked salmon is removed
from the oven, resting for a few minutes on the baking
sheet to allow the juices to settle.
Action: Remove the salmon from the oven and let it
rest for a few minutes before serving. *
```

