# OpenReview forum: "Long-horizon Visual Instruction Generation with Logic and Attribute Self-reflection"
_ICLR.cc/2025/Conference — ICLR 2025 Poster_

### Official Review · Reviewer_UNJf · 2024-10-31

**Soundness:** 3
**Presentation:** 3
**Contribution:** 2
**Rating:** 6
**Confidence:** 4

**Summary:**

This paper proposed a training-free framework for generating long-horizon visual instructions with logic and attribute self-reflection. It drafts consistent images for each step, using historical prompts and visual memory, and corrects errors with image editing tools. Experiments show it produces more comprehensive visual instructions than baseline methods.

**Strengths:**

1.  The training-free framework and self-reflection mechanism of LIGER provide a novel approach to visual instruction generation for long-horizon tasks.
2.  The writing is clear and well-structured, making the concepts easy to understand.
3. The manually curated benchmark tests effectively demonstrate the advantages of the images generated by LIGER in terms of comprehensibility.

**Weaknesses:**

1. In Automatic evaluation, the authors didn't evaluate the quality of the generated images, only assessing alignment and consistency.
2. In this paper, the benchmark's narrow focus on cooking might not capture the full spectrum of complexities and variations present in long-horizon tasks across different industries or activities. It would be beneficial for the authors to provide more details on the types of tasks included in the benchmark.

**Questions:**

1. How does LIGER's self-reflection mechanism ensure that the identification and correction of errors in images are accurate and error-free? Is there a possibility of over-correction or failure to recognize certain types of errors? How you balance Over-consistent and Identity inconsistent?
2. How much time consumption does LIGER introduce?
3. Given the benchmark's focus on cooking tasks, how does LIGER address the generalization to other long-horizon tasks, and are there plans to broaden the benchmark's scope? Like engineering and sports?
4. Is there a specific template or structure that the benchmark follows for the steps involved in the tasks?

---

> ### Author Response · Authors · 2024-11-21
> **Rebuttal by Authors**
>
> We thank the reviewer for the valuable questions. A point-to-point reply is provided below:
> ***
> > **W1:** Image quality evaluation.
>
> The core evaluation criterion for this task is whether the generated images assist human comprehension, while the image quality is indeed important. However, it is hard to evaluate using conventional metrics since there are no ground truth images.
>
> To evaluate the image quality, we add GPT evaluation and user study. Specifically, we prompt the GPT4o model to rate the quality of an image on a scale of 1 to 5 (5 is the best) and test the whole dataset. For the user study, we recruited 5 users, let them view 50 images from each method, and select the best image from the three methods. The win rate and GPT score are reported in Table 9 of the appendix. Note that LIGER is a training-free method, meaning the image quality also depends on the pre-trained tools.
>
> > **W2:** Dataset scenarios.
>
> The proposed dataset mainly focuses on the cooking domain but also involves daily housework. We showcase some qualitative results on other scopes in Figure 10. The reason for using the cooking scenario is that:
>
> Firstly, cooking scenarios present significant challenges representative of long-horizon tasks. Long-horizon tasks usually involve **multiple sub-procedures**, requiring the instruction images to show **scene diversity in different stages**. The cooking procedure reflects this challenge as it usually involves various stages like preparing, cooking, serving, etc. Moreover, the **object attribute in long-horizon tasks changes as the task progresses**, which is also presented in cooking scenarios. For instance, instructions for cooking steak should start with a raw steak and end up cooked. Therefore, testing in the cooking domain is meaningful and representative. Current methods struggle to deal with these challenges in long-horizon tasks.
>
> Secondly, we acknowledge some limitations. Since LIGER is training-free, the generation ability depends on the pre-trained diffusion model. We find the generation quality of fine-grained actions and part of a complex structure unsatisfying.
>
> > **Q1:** Error cases.
>
> The identification and correction are not error-free. There could also be cases of over-correction, though rarely. We add the bad case discussion in Figure 9 of the appendix. To mitigate the errors, we allow the error detector to finish its trial before the referee agent decides whether the edited image is better than the draft image. If the edited image is not ideal, we pick the draft image as the final output.
>
> Over-consistent and identity inconsistent are different levels of errors, over-consistent is a global level error where the images of two steps should not look alike. In contrast, identity inconsistency is an object-level error, meaning the object in two steps should look the same, but not. Over-consistent is a more noticeable error, so the error detector **first assesses over-consistent error**. If an over-consistent error exists, the image is directly regenerated. If no over-consistent error is found, then the error detector starts to inspect inconsistent errors.
>
> > **Q2:** Inference time consumption
>
> We conducted a speed test over 50 randomly selected tasks using a locally deployed multi-modal large language model. LIGER takes around 120 seconds to generate instructions for a 10-step task while the frozen stable diffusion model takes around 60 seconds on a single A100 GPU.
>
> > **Q3:** Dataset scope.
>
> LIGER is a high-level method for long-horizon tasks. It addresses common issues in long-horizon tasks, making it widely applicable. Cooking scenarios are representative since they involve the challenges aforementioned. We also show some additional results in Figure 10 of the appendix.
>
> In the future, we plan to broaden the benchmark and explore more scenarios. However,  we acknowledge certain challenges when testing on other scopes. Firstly, there are tasks hard to illustrate by images like "How to teach your child?" Second, since LIGER is a training-free manner, the generation quality depends on the pre-trained diffusion model. The pre-trained diffusion models struggle to generate fine-grained actions like sewing the knit or building the house using different tools. Despite these challenges, we are optimistic that LIGER will benefit from future advancements.
>
> > **Q4:** Step generation template.
>
> We do not have a specific template for the GPT model to generate steps. The motivation is to test whether LIGER is robust toward instructions in different styles.
> ***
> We hope the above discussions address the concern and are willing to have more discussions.

---

> > ### Comment · Reviewer_UNJf · 2024-11-27
> >
> > Thank you for your response. Most of my concerns are resolved, so I've decided to raise my score in appreciation to 6.

---

> > > ### Author Response · Authors · 2024-11-27
> > >
> > > Thank you so much for the positive feedback!

---

### Official Review · Reviewer_P3KH · 2024-11-03

**Soundness:** 2
**Presentation:** 3
**Contribution:** 3
**Rating:** 6
**Confidence:** 3

**Summary:**

This paper focuses on long-horizon text-to-image generation tasks and proposes a train-free framework. Long-horizon text-to-image tasks face two main challenges: temporal consistency and the accumulation of attribute errors. To address this, the work injects historical text descriptions and visual tokens into a Diffusion Model, then leverages GPT-4 and the segmentation large model LISA to detect and locate errors in the images. Image editing tools are subsequently employed to correct these errors. Meanwhile, the framework uses DDIM inversion to obtain the features of the edited images. Additionally, the paper introduces an evaluation dataset of over 500 samples to assess the effectiveness of long-horizon image generation.

**Strengths:**

- The paper is well-structured and smoothly written, making it easy for readers to understand.
- The figures and tables in the paper are well-designed, making them easy to understand.
- The motivation is well-defined. To address the temporal consistency issue in long-horizon visual instruction generation, it proposes injecting historical text and visual information into the diffusion model. To tackle the attribute error problem, it introduces a method of using MLLMs to detect errors and then calling an agent to correct them, which demonstrates a certain level of novelty.

**Weaknesses:**

- This paper only validates its effectiveness on a self-proposed evaluation dataset, which is somewhat unfair. It is recommended to find more suitable and fair evaluation datasets for verification.
- The evaluation dataset used in this paper is limited to cooking scenarios, lacks generality, and is relatively small, with only about 500 samples.

**Questions:**

- In Section 3.1, "Visual Memory Sharing" uses an attention module to inject historical visual information into the Diffusion model. However, the paper claims that the method is train-free, so where are the weights for this attention module loaded from? If this attention module is not trained on the specific task, can it really be used in a zero-shot manner? I have doubts about its performance.
- The symbol $O_i$ in Eq.3 may lack an explanation, which might leave readers wondering how $O_i$ is used in the Diffusion model.
- In Section 3.2, it is recommended to briefly explain how the image editing tools mentioned (such as DragonDiffusion, SD inpainting Rombach, LAMA, etc.) are used in this method.

---

> ### Author Response · Authors · 2024-11-21
> **Rebuttal by Authors (1/2)**
>
> We thank the reviewer for the question, here is a detailed discussion:
> ***
> > **W1:** *Comparison fairness*.
>
> The proposed dataset is the first dataset curated for the long-horizon visual instruction generation task. The textual tasks are sourced from the widely-used benchmarks and **LLM creates the steps without any specific template, ensuring fairness**. The automatically curated tasks also assess the robustness of different methods given diverse instructions. Human annotators are only responsible for writing their comprehension toward these automatically generated instructions. Furthermore, **all methods are evaluated using the same metrics**. Overall, the comparison conducted on this dataset is fair and reasonable.
>
> The dataset highlights the complexities of this task, i.e. generating images comprehensible with **object state changes, identity consistency, and scene diversity**. These features have not been examined in existing benchmarks. The proposed dataset includes a variety of tasks of different lengths and randomly generated textual instructions, providing a comprehensive test of different methods.
>
> The proposed framework LIGER is also applicable to other datasets. We qualitatively show the generation results on the recipeplan dataset against baseline methods in Figure 11 of the appendix. For the shorter tasks, LIGER still generates illustrative instructions.
>
> > **W2:** *Small dataset restricted to cooking domain.*
>
> We chose cooking scenarios because they encapsulate the key challenges of visual instruction generation:
>
> **(1) Multiple scene setting:** Cooking involves various stages, e.g. preparation, cooking, and serving, each occurring in different scenes. Good instruction should have various scenes under different stages, showing scene transitions.
>
> **(2) Various object and attribute status:** The cooking process usually has various ingredients in different states. As the task progresses, the image should reflect these state changes. For instance, when baking chicken wings, the chicken should be initially raw, while end up cooked.
>
> **(3) Logically coherent between steps:** Cooking scenarios require logical coherent between consecutive steps. Many incremental instructions like "add salt and pepper" should follow the previous step image. On the other hand, there are also logical independent steps that should maintain visual divergence. For example, after preparing vegetables, we need to heat oil in a pan, these two steps should not look alike. This requirement is particularly important in long-horizon tasks.
>
> Having the above challenges, evaluating the cooking scenario effectively reflects the ability of different methods. Current methods also mainly focus on the cooking domain, yet they struggle with the above challenges.
>
> Another reason is that **many tasks are not suitable for image instructions** such as "How to spend your leisure time?" or "How to become a professor". In contrast, visual instructions for cooking scenarios are easier to understand. For instance, merely using the text "wait for a steak well-done", users can not tell whether the steak is well-cooked. Instead, showing a picture of a well-cooked steak clarifies the situation.
>
> LIGER is also suitable for tasks in other daily tasks. Additional qualitative results are shown in Figure 10 of the appendix. Since LIGER is a training-free framework, the generation quality depends on the diffusion model. Some detailed fine actions are difficult to generate due to the limited ability of the pre-trained diffusion model.
>
> The proposed dataset contains 569 long-horizon tasks, involving generating **over 5500 images in one trial**, which is substantial. Curating the dataset also cost some time since the annotators had to annotate over these 5500 instructions.

---

> ### Author Response · Authors · 2024-11-21
> **Rebuttal by Authors (2/2)**
>
> > **Q1:** *Detail of the visual memory mechanism.*
>
> Visual memory sharing is a training-free mechanism that utilizes the visual feature in the previous step to guide the current step generation. Specifically, during the generation of each step, we inject the previous step image visual feature tokens into the **attention operation of the pre-trained UNet**, making the network aware of the previous step. Note that the attention mechanism is inherently designed in the latent diffusion models. We simply load the weight of the pre-trained SDXL model. This technique is also used in other works for identity-keeping and has been proven to be effective [1][2]. We also conducted qualitative ablation and quantitative ablation in Table 2 and Figure 6 of the manuscript.
>
> > **Q2:**  *$O_i$ in Eq3*
>
> $O_i$ in Eq 3 is used in the UNet $U$ in the future deeper layers. We further clarify it in the paper.
>
> > **Q3:**  *Usage of the tools*
>
> The input of DragonDiffusion involves two masks in the previous step image and the current step image respectively. The masks are generated by the LISA model using the text generated by the error detector. SD inpainting takes the new object description generated by the error detector and a mask of the wrong objects as input. LAMA takes a mask of the redundant object as input. We will further clarify this in the paper.
>
> [1] Training-Free Consistent Text-to-Image Generation
>
> [2] Consistent self-attention for long-range image and video generation
> ***
> We hope the above discussions address your concern. We are looking forward to having more discussions about the paper.

---

> > ### Comment · Reviewer_P3KH · 2024-11-27
> >
> > Thank you for the detailed response and additional experiments. They provide helpful clarification, I will keep my positive rating.
> >
> > I wish the authors the best with this work moving forward.

---

> > > ### Author Response · Authors · 2024-11-27
> > >
> > > We sincerely thank you for your appreciation!

---

### Official Review · Reviewer_nhsY · 2024-11-03

**Soundness:** 2
**Presentation:** 3
**Contribution:** 2
**Rating:** 6
**Confidence:** 4

**Summary:**

The paper proposes a method for visual story generation that is completely training-free but consists of many steps. The main parts of the method include keeping a historical prompt and a visual memory for consistency and self-reflection for refinement. The method shows improved results over the previous methods.

**Strengths:**

1. the paper is written well.
2. the method section reads clearly.
3. the paper shows improved results over the baselines.

**Weaknesses:**

1. the method has a lot of moving parts. i would have liked to see some error analysis regarding how does the approach work if one of the components makes a mistake. for example, what happens if gpt-4o misses some details?
2. the current work is overly reliant on the closed-source gpt-4o. there are also plenty of open source models available. some ablation on using open source vlms could be useful and beneficial for the community.
3. error bars are not reported.
4. how expensive is this approach. if the sequence length is too long of the tasks. would that mean that we will need to store a lot of 'visual memory'? some discussion on this could have been helpful.

**Questions:**

i have some serious concerns about the motivation of this task. what can be the differences of this particular task with semantically consistent video generation for a scene?

please also look at the weaknesses. overall i like the paper. but if these concerns are addressed i can update my score.

---

> ### Author Response · Authors · 2024-11-21
> **Rebuttal by Authors**
>
> We provide a discussion on the valuable questions posed by the reviewer.
> ***
> > **W1:**  *Error analysis*.
>
> We show some cases in Figure 9 in the appendix. In practice, LIGER employs a referee agent to judge the mistakes in the edited image. Figure 9 shows two types of errors in the edited images, i.e. **Reasoning error and generation error**.
>
> For instance, in the reasoning error case, the precious step involves whisking eggs into the batter. The current step is to add vanilla extract. However, the error detector mistakenly assumes the egg should still be visible in the current step, which is not the case after whisking. The referee agent finds the error and logic error and keeps the draft image as the final output.
>
> Generation error, on the other hand, is attributed to failures of the editing tools or the location tool. In the example, the balls are mistakenly removed and also the quality of the generated image is unsatisfactory. The referee agent, considering image quality, identifies the mistake and picks the draft image as the final output.
>
> Having this rollout-and-compare manner, LIGER is robust toward the errors. It is also worth mentioning that there such errors are infrequent.
>
> > **W2:** *Robustness toward MLLM*
>
> We conducted an ablation study substituting the GPT4-O model with open-source models Pixtral-12B and QwenVL-7B. Qualitative results in Table 6 reveal a performance degradation using the Pixtral-12B model, nevertheless, the self-reflection mechanism maintains effectiveness. MLLM capability influences the performance, as demonstrated by the inferior performance of QwenVL-7B compared to Pixtral-12B. We empirically find that the reasoning and image comprehension abilities are limited when using open-source models.
>
> > **W3:** *Error bars*.
>
> We performed another trial over the 569 tasks, and the automatic evaluation results are shown in Table 7. The variance is relatively small. To further examine the variance, we randomly selected 50 tasks and ran 5 separate trials. Due to time and budget constraints, these additional trials were limited to the subset. Results in Table 8 show that the variance is also small and all the trials consistently outperform baseline methods.
>
> > **W4:** *API cost and visual memory.*
>
> A trial on the 569 tasks costs around 200 dollars in total, 0.035 dollars per image.
>
> The task length does not affect the visual memory amount, since we update the visual memory every step and only store the memory for the current step. The visual memory is stored in a ''slicing window'' manner and invariant with step length.
>
> > **Q1:** *Motivation of the task and difference with consistent video generation*
>
> The motivation of the task can be summarized as:
>
> (1)  We identify challenges in the task of visual instruction generation, i.e. showing **object state changes, object consistency and scene diversity** between steps. The challenges are exemplified especially in long-horizon tasks.
>
> (2) Visual instruction generation has great **real-world application potential**. Visual instructions are crucial for human learning as they are intuitive. With the rise of social media, users often refer to image blog posts to quickly grab key information and the status of the tasks on their mobile devices. Generating image instructions meets the needs of users. Furthermore, this task can also be integrated with large language models, enabling them to respond to user queries with illustrative processes.
>
> (3) Visual instruction generation lays the foundation for future work. **The images generated by LIGER could serve as highlight frames to guide video generation**.  Moreover, generating image instructions potentially assists other scenarios such as embodied agent planning. Agents or robots can adapt to new tasks using the generated instructions.
>
> Regarding the difference between image instruction generation and semantically consistent video generation, we find that existing **open-source text-to-video generation models primarily focus on single object scene consistency, and struggle to directly generate long videos from text that exhibit reasonable scene diversity and state transitions.** Image instruction generation requires object identity consistency, scene diversity, and object state changes to make images easy to understand. Moreover, text-to-video generation can be broken down into text-to-image generation and image-to-video generation. Images generated by LIGER can be used as the highlight frames for the videos, aiding video generation by facilitating the text-to-image generation phase.
> ***
> We hope the discussions address your concerns. We are happy to have more discussions with you. Thank you again for your valuable opinions.

---

> > ### Comment · Reviewer_nhsY · 2024-11-25
> >
> > Thank you for the clarifications. I will raise my score to 6. Good luck!
> >
> > Best,

---

> > > ### Author Response · Authors · 2024-11-25
> > >
> > > We sincerely thank you for your appreciation!

---

### Official Review · Reviewer_fPFq · 2024-11-04

**Soundness:** 3
**Presentation:** 3
**Contribution:** 2
**Rating:** 6
**Confidence:** 4

**Summary:**

This manuscript introduces an innovative task named"Visual Instructions for Long-Horizon" which aims to generate a series of continuously aligned visual images corresponding to textual instructions. The manuscript proposes four self-reflection methods that leverage both visual and historical prompts. To prevent cumulative deviation and help generate along the correct trajectory, an approach termed Inversion-Based Visual Memory Calibration is proposed. The proposed method is noteworthy for its approach to addressing attribute errors and inconsistencies by utilizing existing mLLM tools.

**Strengths:**

1.	The proposed task and solution are indeed novel. Ablation experiments validate the contributions of the various proposed modules, including the visual prompt, historical prompt, self-reflection mechanism, and inversion-based memory calibration.
2.	Numerous qualitative experiments demonstrate that the proposed tool-based self-reflection method maintains alignment with textual instructions, ensuring that the generated images adhere to contextual logic, thereby validating the overall efficacy of the approach.

**Weaknesses:**

1.	This manuscript would benefit from a more comprehensive comparative analysis. I recommend including comparisons with additional train-free methods, such as [1] and [2], as well as approaches that share similar concepts, like [3], and non-tool-based methods, such as [4].
2.	Furthermore, not all metrics proposed by the authors are original, and the use of multimodal large language model (LLM) reasoning to evaluate images also not original. It is advisable to revise the relevant statements in the contributions section to reflect this accurately.
3.	While the research method is innovative, the reliance on several pre-designed strategies and external models for image refinement may not be particularly efficient in practical applications. This could be seen as a limitation of the manuscript and the authors are encouraged to provide an appropriate discussion of this issue.

+ [1] Coherent Zero-Shot Visual Instruction Generation
+ [2] Training-Free Consistent Text-to-Image Generation
+ [3] Consistent self-attention for long-range image and video generation
+ [4] StoryDiffusion: Consistent Self-Attention for Long-Range Image and Video Generation

**Questions:**

Why is there no evaluation of the effect of “+self-reflection“ alone in the ablation experiments? It appears to be directly combined with memory calibration in the final method without independent verification.

---

> ### Author Response · Authors · 2024-11-21
> **Rebuttal by Authors**
>
> We thank the valuable suggestion and appreciation from the reviewer. Below we provide a point-to-point reply.
> ***
>
> > **W1:** *Related work comparison*.
>
> We have added a qualitative comparison between ConsiStory, Storydiffusion, and the proposed LIGER in Appendix A2 of the manuscript pdf file. We have included the papers you mentioned in the related work section and discussed the differences.
>
> Specifically, Long-horizon Visual instruction generation requires not only object consistency but also scene diversity between images. As shown in the figure, ConsiStory, and Story diffusion **lack scene diversity**, also the **object state is static**. Moreover, the two methods also need to **identify the subject concept expression manually**, which is not required by LIGER.
> In the image example, pork shows an appearance change from raw to well-done. The reason is that previous methods only maintain consistency through sharing hidden states in the UNet, while LIGER leverages the reasoning ability of large language models.
>
> > **W2:** *Statements on evaluation metrics*.
>
> We have modified the statements in the contribution section of the introduction. We clarify that the **testing goals of these metrics are different**.
>
> (1) First, the CLIP-Score is used to test the alignment between the generated images and human perceptions. The human perceptions are represented by hand-written text annotations.
>
> (2) The DINO ratio score is designed to evaluate both the coherence between related steps and the distinction between unrelated steps by measuring image similarity and divergence respectively.
>
> (3) Moreover, the BERT score evaluates image illustrativeness through a modality transfer test which is innovative and different from previous metrics.
>
> In terms of the GPT evaluation, we further evaluate the generated instructions from the perspective of Large language model comprehension. Overall, the evaluation metrics reveal different challenges and provide a thorough comparison.
>
> > **W3:** *Inference efficiency*
>
> Indeed, incorporating various tools inevitably raises inference time. Our speed test reveals that vanilla SDXL can generate images for a 10-step task in about 60 seconds while LIGER takes about 120 seconds. Despite LIGER being slower than the vanilla diffusion model, it exhibits enhanced generation quality. Moreover, LIGER is still significantly faster than the human design process, which has real-world potential to generate comprehensive images. We include these discussions in the limitation section of the appendix.
>
> > **Q1:** *Self-reflection mechanism ablation.*
>
> We conduct further ablation study of SDXL+self-reflection (namely SDXL+R), which is essentially LIGER without visual memory and history prompt. Quantitative results are shown in Table 4 in the appendix of the manuscript. A performance improvement over vanilla SDXL is observed, demonstrating its effectiveness.
>
> To dive into the effectiveness of the self-reflection mechanism, we also test the performance of SDXL+V+R and SDXL+H+R. However, due to the time limit and budget, we provide the result on a 100-task subset in Table 5 of the manuscript. Results show that the self-reflection mechanism consistently improves performance.
> ***
> We hope the response addresses the concern. We are willing to have discussions if there are still questions.

---

> ### Author Response · Authors · 2024-12-01
>
> Dear Reviewer fPFq,
>
> As the deadline of the discussion period was extended, we conducted the additional ablation experiment on the whole dataset.
>
> The results of the whole 569 tasks are shown in the table below.
>
> | Method | CLIP-Score $\uparrow$ | DINO-Score $\downarrow$ | BERT-Score $\uparrow$|
> | -------- |  --- |  --- |  --- |
> | SDXL | 2.5837 | 0.8516 | 0.8699 |
> | SDXL+V | 2.6251 | 0.8239 | 0.8719|
> | SDXL+H | 2.6842 | 0.8224 | 0.8707|
> | SDXL+R |  2.7270 | 0.7346 | 0.8732|
> | SDXL+H+V | 2.7168 | 0.7459 | 0.8721|
> | SDXL+V+R | 2.7428 | 0.7053 | 0.8734|
> | SDXL+H+R | 2.7440 | 0.6653 | 0.8740 |
> | LIGER | 2.7555 | 0.6338 | 0.8743|
>
> If you have any other questions or concerns, we are always here to do our best to answer them. We look forward to continuing our discussion with you.
>
> Best regards,
>
> Authors

---

> ### Author Response · Authors · 2024-12-03
> **A Kind Reminder Regarding Our Rebuttal.**
>
> Dear Reviewer fPFq,
>
> The discussion period will close in eight hours. We understand that you may be busy, but we are eager to hear back from you at your earliest convenience.
>
> We highly value your feedback and thank you again for your patience and expert opinion during the review process.
>
>
> Best regards,
>
> Authors

---

### Author Response · Authors · 2024-11-22
**Manuscript Update Information**

We thank the reviewers for their insightful and professional suggestions. Based on the comments, we updated the manuscript in the following sections:

- **Quantitative experiments** (Appendix A.1.)

    1. Additional ablation on the self-reflection mechanism in Table 4 and Table 5.

    2. Ablation on MLLM model in Table 6.

    3. Variance test in Table 7. and Table 8.

    4. Image quality evaluation in Table 9.

- **Qualitative experiments** (Appendix A.2.)

  1. Comparison with Consistory and StoryDiffusion in Figure 8.

  2. Error case analysis in Figure 9.

  3. Qualitative results on other scenarios in Figure 10.

  4. Recipeplan dataset example in Figure 11.


- **Text Revision**

  1. We modified the statement of evaluation metrics.

  2. Adding an Explanation of how the output in Eq 3. is used. The inputs of the editing tools are clarified.

  3. Limitation discussions are included in the appendix A.3.

All the modifications are marked blue.

---

### Author Response · Authors · 2024-11-25
**Gentle Reminder Regarding our Rebuttal**

We sincerely thank reviewers for their professional and detailed reviews.

As the discussion period deadline approaches, we are looking forward to receiving feedback on our rebuttal, so we can engage in further discussions and refine our work.

We understand the reviewers may be under a large workload, and we are grateful for their time and patience.

Best regards, authors.

---

### Meta-Review · Area_Chair_429T · 2024-12-18

**Metareview:**

The paper presents LIGER, a novel training-free framework for generating long-horizon visual instructions which utilizes historical prompts and visual memory. The framework is designed to improve the consistency and accuracy of instructional outputs. According to the reviewers (UNJf, nhsY), LIGER shows promising results on a new benchmark dataset, with particular emphasis on cooking tasks.

The strengths of the paper include the innovative training-free approach and the self-reflection mechanism of LIGER, which are recognized as significant contributions to the field (UNJf, P3KH). Additionally, the paper is well-structured and clearly written, and the introduction of a new benchmark dataset is commended for providing a solid foundation for evaluation (P3KH, UNJf, nhsY, fPFq).

However, the weaknesses lie in the limited scope of the dataset, focusing primarily on cooking scenarios which may affect the generalizability of the findings (P3KH, UNJf). The reliance on proprietary models like GPT-4 also raises concerns about accessibility and replicability (nhsY, fPFq). A notable oversight is the absence of a detailed error analysis and robustness testing (nhsY, UNJf).

The decision to accept rests on the novelty and effectiveness of LIGER, despite the raised issues. The authors' thorough responses during the rebuttal period have been considered to balance out the concerns about generality and reliance on closed-source models.

**Additional Comments On Reviewer Discussion:**

During the review discussion, concerns were raised about the depth of analysis of alternative methods and baseline comparisons (fPFq), the narrow focus of the benchmark dataset (P3KH, UNJf), and the lack of error analysis (nhsY).

The authors addressed these by providing additional comparisons with alternative methods, expanding upon their benchmark dataset to improve its scope (fPFq, P3KH), and including error analysis and robustness tests in their rebuttal. These additions have alleviated some concerns, though the performance impact of using open-source models instead of proprietary ones is still unclear (nhsY).

In weighing these concerns, the authors' comprehensive rebuttal and supplementary experiments have been taken into account. The originality and responsiveness to feedback have been pivotal in the decision-making process.

---

### Decision · Program_Chairs · 2025-01-22

Accept (Poster)